# New Insights into the Antioxidant Compounds of Achenes and Sprouted Buckwheat Cultivated in the Republic of Moldova

Caterina Dumitru [1], Rodica Mihaela Dinica [2,*], Gabriela-Elena Bahrim [3], Camelia Vizireanu [3], Liliana Baroiu [1,*], Alina Viorica Iancu [1,*] and Miruna Draganescu [1]

[1] Faculty of Medicine and Pharmacy, "Dunarea de Jos" University from Galati, 47 Domneasca Str., 800008 Galati, Romania; caterina.dumitru@ugal.ro (C.D.); draganescumiruna@yahoo.com (M.D.)
[2] Department of Chemistry, Physics and Environment, "Dunarea de Jos" University from Galati, 111 Domneasca Str., 800201 Galati, Romania
[3] Faculty of Food Science and Engineering, "Dunarea de Jos" University from Galati, 111 Domneasca Str., 800201 Galati, Romania; gabriela.bahrim@ugal.ro (G.-E.B.); camelia.vizireanu@ugal.ro (C.V.)
* Correspondence: rodica.dinica@ugal.ro (R.M.D.); lilibaroiu@yahoo.com (L.B.); iancualina.2003@yahoo.com (A.V.I.)

**Abstract:** It is well known that both *Fagopyrum esculentum* species, buckwheat achenes and buckwheat germs, contain flavonoids, and that they can be considered functional foods. In the present study we have analyzed the total content of polyphenols (TPC) and flavonoids (TFC), as well as the antioxidant activity of buckwheat ahcenes originating from the Balti region, Republic of Moldova, and also of the buckwheat sprouts over seven days of germination. The content of phytochemicals in achenes and germinated buckwheat after three and seven days was determined by HPLC–MS analysis. Using the Folin–Ciocalteu method, we recorded an increase in TPC of 360%, and of 436% in TFC after seven days of buckwheat ahcene germination. We aimed to investigate the free radical scavenging properties of methanolic extracts from ahcenes and sprouted buckwheat. We identified and quantified flavonoids and lignans such as rutin, orientin, isoorientin, vitexin, quercetin, quercitrin, isorhamnetin, lutein, apigenin, catechins, coumestrol—which have countless beneficial effects on human health—using HPLC–MS. FTIR analysis also revealed the accumulation of phenolic compounds during germination. This is the first study on the identification and quantification of phytochemicals from buckwheat achenes and sprouts from the Balti region of the Republic of Moldova.

**Keywords:** buckwheat seeds; germinated buckwheat; flavonoids; lignans; rutin; vitexin; FTIR spectrum; functional food

## 1. Introduction

Through evolution, plants have developed the ability to produce a wide range of metabolites with roles in their own physiological processes as well as in combating biotic and abiotic stress, and metal toxicity.

Polyphenols, with over 8000 representatives, are secondary metabolites that appear in plants as a reaction to the action of ultraviolet rays, and to the aggression of pathogens [1]. Depending on their chemical structure, they can be classified as flavonoids or non-flavonoids. Flavonoids are polyphenolic compounds that have two aromatic rings (commonly denoted as A and B) linked by a three-carbon chain. The connecting chain between the two rings (A and B) is made by a heterocyclic ring (C)—except for the chalcones, where the chain is linear. (Figure 1). The main subclasses of these C6–C3–C6 compounds are the flavones, flavonols, flavan-3-ethanols, isoflavones, flavanones and anthocyanidins.

**Flavonols**

| Flavonols | R1 | R2 | R3 |
|---|---|---|---|
| isorhamnetin | H | CH3 | H |
| kaempferol | H | H | H |
| quercetrin | OH | H | rhamnose |
| quercetin | OH | H | H |
| rutin | OH | H | rutinose |

**Flavones**

| Flavones | R1 | R2 | R3 | R4 | R5 | R6 |
|---|---|---|---|---|---|---|
| vitexin | H | OH | OH | H | OH | glucose |
| isoorientin | OH | OH | OH | glucose | OH | H |
| orientin | OH | OH | OH | H | OH | glucose |
| luteolin | OH | OH | OH | H | OH | H |
| apigenin | H | OH | OH | H | OH | H |

**Izoflavones**

| Isoflavones | R1 | R2 | R3 |
|---|---|---|---|
| daidzein | OH | H | OH |
| genistein | OH | OH | OH |

**Flavanones**

| Flavanones | R1 | R2 | R3 | R4 |
|---|---|---|---|---|
| hesperidin | OH | OCH3 | rutinose | OH |
| hesperetin | OH | OCH3 | OH | OH |
| naringenin | H | OH | OH | OH |
| naringin | OH | H | neohesperidose | OH |

**Anthocyanidins**

| Anthocyanidins | R1 | R2 |
|---|---|---|
| pelargonidin | H | H |
| malvidin | OCH3 | OCH3 |
| delfinidin | OH | OH |
| cyanidin | OH | H |

**Flavononols**

| Flavononols | R1 | R2 | R3 | R4 |
|---|---|---|---|---|
| taxifolin | OH | OH | OH | OH |
| aromadendrin | OH | H | OH | OH |

**Flavan-3-ols**

| Flavan-3-ols | R1 | R2 | R3 | R4 | R5 | R6 |
|---|---|---|---|---|---|---|
| catechines | OH | OH | H | OH | OH | OH |
| gallocatechin | OH | OH | OH | OH | OH | OH |
| gallocatechin gallate | OH | OH | OH | OH | OH | gallate |

**Chalcones**

**Flavan**

**Figure 1.** Chemical structures of important flavonoid sub-classes.

The dietary sources of these flavonoids are cereals, fruits and vegetables, and their consumption has been associated with a number of beneficial effects on health. As a result, many studies have focused on the analysis of different plants for the identification and quantification of flavonoids with a role in health promotion. One of these plants, quite well studied in certain geographical areas, is the common buckwheat, *Fagopyrum esculentum*. This species, together with another 1200 species, is part of the Polygonaceae family—with a wide distribution in Europe, in the temperate zone of North America and in Southeast Asia [2], Russia and China. The wild forefather of common buckwheat is *F. esculentum ssp. ancestrale*. It seems that the origins of this pseudocereal were in China, about 5000–6000 years ago. It has been suggested that buckwheat arrived in Europe between 4000 and 2800 BC, and from there, through immigrants to North America. [3].

Functional foods are products that contain various biologically active compounds, which when consumed in the diet, contribute to maintaining the optimal state of physical, mental and general health of the population. As functional foods are part of the daily diet, their effects are long-lasting. For example, foods containing probiotics and/or prebiotics and/or antioxidants can be considered functional foods. Buckwheat is a functional food because it is a rich source of protein, with adequate ratios of amino acids, fiber, minerals (especially zinc, copper, and magnesium) and vitamins (B1, B2, B3 and B6). It has significant amounts of phenolic compounds—especially flavonoids [2–4]—with antioxidant, anti-inflammatory and antitumor actions. Research has shown that buckwheat has a much higher antioxidant capacity than oats, barley [5] wheat [6] and other pseudocereals [5,7,8], due to the higher content of polyphenolic compounds. The lack of gluten makes it more attractive for people with celiac disease or gluten intolerance. As buckwheat achenes and buds are rich sources of polyphenolic compounds, they began to be used in the manufacture of bread, noodles, pasta and functional drinks [9]. Buckwheat is also grown to support honey production [10] and is considered one of the best plants for obtaining honey. For many years, the therapeutic value of buckwheat honey has been known, which contains,

in addition to a mixture of sugars, significant levels of minerals, vitamins, antioxidants, and other biologically active components [11] specific to the honey plant.

For these reasons, in recent years, buckwheat has been considered as a healthier alternative to cereals.

Traditionally, in many countries, sprouts are part of their culinary history. From a physiological point of view, sprouting of achenes is defined as the appearance of the root in the seed, and is the process by which seeds grow, giving rise to the new plant. This is the only procedure that achieves a significant increase in the nutritional value of food by increasing the bioavailability of nutrients, and the content of vitamins, bioelements and other biologically active substances [12].

After the 1990s, epidemiological studies have highlighted the positive role of antioxidants in sprouts on oxidative stress, with beneficial implications in most conditions. As a result, the popularity of sprout consumption in Western countries has increased due to the high consumer demand for healthy foods.

The beneficial effects of buckwheat and sprouted buckwheat on health—lowering cholesterol [13] regulating hypertension [14] inhibiting tumor growth [15] and controlling diabetes [16] have been highlighted. Currently, buckwheat is used not only to improve the nutritional value of food, but also to produce food with a beneficial impact on human health [17], or to obtain preparations in the pharmaceutical or cosmetic industries [18,19]. One characteristic of buckwheat is the rather high content of rutin and O-glycosyl flavonols [20]. Additionally, significant amounts of vitexin, orientin, isoorientin and quercetin have been identified in both its seeds and sprouts [21]. The flavonoid content differs both depending on the different parts of the plant and the time of harvest, as well as the geographical region. As sprouts contain a greater number of flavonoids than seeds, their antioxidant activity is also higher. For this reason, buckwheat sprouts have been and are still being studied to identify their specific antioxidant profile in certain geographical areas. The evolution of antioxidants during germination can also be analyzed by infrared spectroscopy. This is a method based on the vibrations of the atoms of a molecule. Fourier transform infrared spectroscopy (FTIR) is a rapid analytical method that allows continuous monitoring of the baseline spectral line, thus achieving a distinct molecular fingerprint [22].

Over the next 5–6 years, the worldwide pharmaceutical market is expected to grow by USD 135.4 billion. It is estimated that by 2025, the functional food and beverage industry segment will grow by 6.1% (US) and 10.7% (China) [23].

From an economic point of view, it is necessary to know the phytotherapeutic potential of plants in a certain geographical region because the content and nature of buckwheat polyphenolic compounds vary depending on the species, variety, environmental conditions and harvesting conditions [5,24,25]. The availability of a large number of antioxidants in buckwheat seeds, knowledge of the dynamics of antioxidant compounds in germinated buckwheat and short harvest times are significant factors that contribute to increasing buckwheat's market share.

The aim of this study is to analyze the antioxidant properties of buckwheat achenes as well as sprouted buckwheat seeds. This study is the first that, using HPLC–MS and ATR-FTIR methods, performs an analysis of flavonoid compounds present in buckwheat and buckwheat seeds sprouted after three and seven days, in the Balti region, Republic of Moldova.

## 2. Materials and Methods

### 2.1. Buckwheat Materials

The analyzed material components were the achenes from the selected cultivars of Sahalin buckwheat (*Fallopia sachalinense Fr. Schmidt*) Gigant variety, according to the Catalog of Plant Varieties for the year 2021 [26].

### 2.2. Chemicals, Reagent and Materials

All standards, with a purity between 95–99% were purchased from Sigma Aldrich GmbH (Steinheim, Germany). The organic solvents methanol and acetonitrile—HPLC grade, formic acid (98%), Tris (hydroxymethyl) aminomethane, acetic acid, ammonium acetate, ammonia and ultrapure water were purchased from Merck (Romania). Solid phase extraction (SPE) StrataX 200 mg 6m/L cartridges were purchased from Phenomenex (Romania). Absolute ethanol, Folin–Ciocalteu phenol reagent and $AlCl_3$ were purchased from Merck Romania. All other chemicals and reagents were of analytical grade

### 2.3. The Sprouting

The sprouting was performed in an EasyGreen automatic machine (USA). Buckwheat achenes were washed several times with water and then kept in clean water for 2 h. They were then placed in the machine tray, in a single layer, with the temperature being maintained at 23–25 °C. For the first 3 days, the buckwheat achenes were washed at 6 h intervals. Starting on the 4th day, the achenes were sprayed with water at intervals of 4 h for 3 min. Samples were taken on the third and seventh day (Figure 2) and dried at 40 °C. After drying, the samples were finely ground in a vacuum. The bags were kept in dry conditions, in the dark, until their usage.

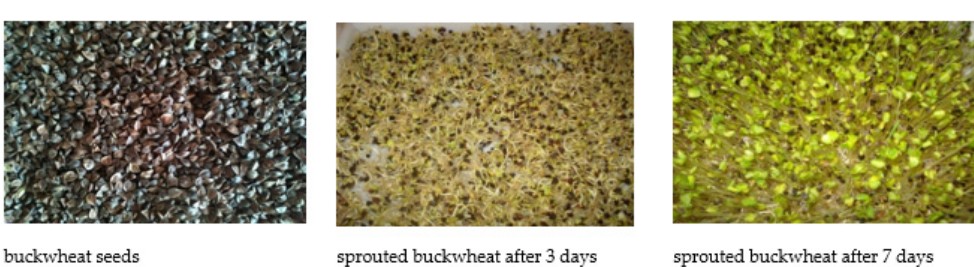

buckwheat seeds    sprouted buckwheat after 3 days    sprouted buckwheat after 7 days

**Figure 2.** Buckwheat achenes and sprouted buckwheat at 1–7 days.

### 2.4. Preparation of Vegetative Material

From the 3 samples, 1 g of plant material was weighed on the analytical balance, over which was added 5 mL of extraction solvent: 1% HCl in methanol. The samples were vortexed for 5 min, sonicated for 30 min, centrifuged for 15 min at 3000 rpm and filtered. The extracts obtained were stored in the freezer and were used to determine the total polyphenols, total flavonoids and antioxidant activity. For HPLC–MS analysis of phenolic compounds, the extract was filtered through a 0.45 µm filter and injected into the HPLC–DAD–MS Agilent 1200 system.

### 2.5. Total Polyphenol Content (TPC) Analysis

The total polyphenol content of the methanolic extracts was determined spectrophotometrically using gallic acid as a standard, according to the method described by Anesini [27] 1.0 mL of the extract was transferred as a triplicate, to tubes containing 5 mL of Folin–Ciocalteu reagent (1:10) and 4.0 mL sodium carbonate solution (7.5% $w/v$). After keeping the tubes at room temperature for 60 min, the absorbance at wavelength λ = 765 nm was read using a UV-Vis spectrophotometer (T80 UV/VIS spectrophotometer) in relation to a control sample. The total polyphenol content was expressed as gallic acid equivalents (GAE) in mg/g dwb. The TPC was determined by interpolation of the sample absorbance against a calibration curve built with gallic acid standards (10, 20, 30, 40, 50 µg/mL in 10% ethanol) and expressed as mg GAE/g dwb, using the calibration curve y = 0.071x + 0.006, $R^2$ = 0.0092. All the experiments were done in triplicate.

### 2.6. Total Flavonoid Content (TFC) Analysis

The measurement of the total flavonoid content in the investigated extracts was performed spectrophotometrically using a method [28] based on the formation of a yellow flavonoid–aluminum complex and its absorbance reading at 430 nm. To 1 mL of methanolic

extract was added 1 mL of 2% $AlCl_3$ solution (2.5% $w/v$), 2 mL of $CH_3COOH$ (10% $w/v$) and ethanol (70% $w/v$) to a volume of 10 mL. After 30 min, the absorbance at wavelength $\lambda = 430$ nm needed to be read. The flavonoid content was expressed as mg quercetin equivalent per 100 g dry-weight basis (mg QE/g dwb) and calculated using the standard curve plotted on the concentration range 10 and 50 µg / mL quercetin: y = 0.0045x + 0.0062, $R^2 = 0.00978$. All the experiments were done in triplicate.

### 2.7. Determination of Antioxidant Activity by the DPPH Method

The determination of antioxidant activity with the DPPH method was performed according to the Brand–Williams protocol [29]. The stock solution was obtained by dissolving 0.025 g of DPPH in 100 mL of methanol. Before the analysis, a 1:10 dilution with methanol from the stock solution was prepared. For analysis, 3.9 mL of DPPH working solution was added to 0.1 mL extract, and after 10 min, the absorbance at wavelength $\lambda = 515$ nm ($A_{sample}$) was read on the T80 UV/VIS spectrophotometer—the color of the solution being proportional to the antioxidant concentration. All the experiments were done in triplicate. The percentage of DPPH radical scavenging activity was calculated by the following equation:

$$(RSA)\% = ((A_0 - A_1)/A_0) \times 100$$

where $A_0$ is the absorbance of the control and $A_1$ is the absorbance of the extractives/standard.

### 2.8. HPLC–MS Analysis of Phenolic Compounds

Phenolic compound analysis was performed using the HPLC–DAD–MS method and the Agilent 1200 HPLC system with a DAD detector linked with an Agilent 6110 quadrupole single MS detector. The separation of phenolic compounds was performed on a chromatographic column: Eclipse XDB C18, dimensions 4.6 mm × 150 mm, 5 µm, Agilent, at the temperature 25 °C. The mobile phase consisted of water + 0.1% acetic acid/acetonitrile (99/1) $v/v$ as eluent A and 100% acetonitrile + 0.1% acetic acid as eluent B. The gradient used over the first 2 min mobile was 95% phase A, 5% B followed by a linear increase of solvent B to 40% until min 18, a linear increase of phase B to 90% over the next 12 min, maintenance for 2 min and then a return to 95% A and 5% B in the next minute, with maintenance for column balancing for 5 min. The total analysis time was 30 min with a mobile phase flow of 0.5 mL/min. The monitored wavelengths were between 240 nm and 370 nm. For MS analysis, positive ionization ESI mode was used. The ionization parameters were: capillary voltage: 3000 V; temperature: 35 °C, nitrogen flow: 8 L/min. The data acquisition was done in full scan mode in the mass range m/z: 100–1000. The identification of phenolic compounds in the samples was obtained by comparing retention times, UV and mass spectra with those of the injected standards. Quantification of the different phytochemicals were based on peak areas of six different concentrations in six replicate assays, being expressed by the linear regression coefficient $R^2$, and calculated as equivalents of representative standard compounds. All contents were expressed as micrograms per g dry weight.

### 2.9. FTIR Analysis

Using the total reflection attenuation (ATR) technique with the Magna-IR Spectrometer 350 (Nicolet Instrument Corporation, USA, ThermoScientific), spectra were recorded from 4000 to 700 cm$^{-1}$ with a spectral resolution of 1 cm$^{-1}$, and scans of the samples were in the region of 4000–650 cm$^{-1}$.

### 2.10. Statistics

The mean values ± standard deviations were calculated using Microsoft Excel 2007. The average value standard deviation is reported. The paired sample t-test it was used to find the significance of difference between pairs of variables. Results with $p$ values $\leq 0.05$ were deemed statistically significant. All experiments were carried out in triplicate.

## 3. Results and Discussions

### 3.1. Evolution of Total Polyphenol Content Assay (TPC) and Total Flavonoid Content Assay (TFC)

It was observed that both TPC and TFC increased significantly ($p < 0.05$, Table 1) after three and seven days of sprouting. Thus, after 3 days of germination, TPC increased by 25%, and after 7 days by 267.5%. Similarly, TFC increased by 37.5% after 3 days of sprouting and by 289.9% after 7 days.

**Table 1.** TPC and TFC, the comparison between our study and the studies which were carried out by Alvarez [30] and Zhang [31].

|  | Buckwheat Seeds | After 3 Days of Sprouting | After 7 Days of Sprouting | Authors |
|---|---|---|---|---|
| TPC, mg GAE/g dwb | $4.91 \pm 0.15$ | $6.14 \pm 0.22$ |  | Our study |
|  | $3.03 \pm 0.09$ | $8.42 \pm 0.44$ | $22.57 \pm 0.71$ | [31] |
|  | $3.23 \pm 0.14$ | $6.70 \pm 0.12$ |  | [30] |
| TFC, mg GAE/g dwb | $4.05 \pm 0.09$ | $5.57 \pm 0.32$ |  | Our study |
|  | $4.17 \pm 0.11$ | $11.69 \pm 0.87$ | $21.72 \pm 0.76$ | [31] |

In the present study, the TPC and TFC values in buckwheat achenes and germinated buckwheat after three days were similar to those reported by Alvarez et al. [30], whereas compared to those reported by Zhang [31] (Table 1), they are observed to be lower. These differences may be because buckwheat sprouting was carried out in a machine and not under natural conditions; the accumulation of polyphenolic compounds is much higher in sunlight.

### 3.2. 2,2-Diphenyl-1-Picrylhydrazyl (DPPH) Radical Scavenging Activity Assay

Buckwheat, especially sprouted buckwheat, is interesting from both a nutritional and pharmaceutical point of view [29] due to its high antioxidant capacity. The antioxidant activity of buckwheat achenes was determined by the DPPH test—DPPH being one of the most stable free radicals. The buckwheat achenes were characterized as having a high radical scavenging activity of 15%. After 3 days it increased to 37%, and after 7 days of sprouting it reached 87%. This growth is explained by the fact that sprouting increased the accumulation of antioxidant compounds.

The antioxidant activity measured by this method was positively correlated, significantly, both with the total content of flavonoids (0.985; $p < 0.05$) and with the free content of polyphenols (0.984; $p < 0.05$).

### 3.3. HPLC–MS Analysis of Phenolic Compounds

Following the HPLC–MS analysis carried out in the present study, the following flavonoid compounds were identified in buckwheat seeds and sprouted buckwheat (Tables 2 and 3).

The flavonoid quantification, which is presented in Table 4, was performed based on the calibration curves drawn for each compound. The analytical method used was performed according to a previous published article [32]. The precision (intra- and inter-day) was the measure of repeatability of the analytical method under the normal operating conditions, and was verified by six determinations of three concentrations of the standards on the same day (intra-day) and on different days (inter-day), which were carried out and expressed as a percent relative standard deviation (RSD%). The relative standard deviation (RSD) ranged from 1.26% to 4.48% for run-to-run precision and from 1.58% to 32.23% for day-to-day precision. The calibration curves were constructed using six different concentrations in six replicate assays and were expressed by the linear regression coefficient $R^2$ (line 318–319). The linearity of the method was demonstrated by the correlation coefficient of the whole calibration curve, which was greater than 0.990.

**Table 2.** Retention time, wavelength characteristic of maximum absorption, and ion monitored in MS for phenolic compounds identified in samples.

| Nr. Peak | Retention Time $t_R$ (min) | $[M-H]^+$ (m/z) | UV $\lambda_{max}$ (nm) | Compound Name |
|---|---|---|---|---|
| 1 | 3.2 | 379, 363 | 280 | Hidroxysecoisolariciresinol (lignan) |
| 2 | 4.7 | 307, 202 | 280 | Epigallocatechin (EGC) |
| 3 | 5.5 | 459, 202 | 280 | Epigallocatechingallat (EGCG) |
| 4 | 7.0 | 459, 202 | 280 | Gallocatechingallat (GCG) |
| 5 | 9.4 | 269, 183 | 240, 300 | Coumestrol *(isoflavonoid)* |
| 6 | 12.1 | 291, 202 | 280 | Catechin |
| 7 | 12.9 | 363, 247, 163 | 280 | Secoisolariciresinol (lignan) |
| 8 | 14.0 | 449, 287 | 270, 350 | Orientin |
| 9 | 14.5 | 449, 287 | 270, 350 | Isoorientin |
| 10 | 15.4 | 433, 271 | 270, 340 | Vitexin |
| 11 | 16.1 | 611, 475, 303 | 250, 360 | Rutin |
| 12 | 16.4 | 493, 475, 317 | 250, 320 | Isorhamnetin glucuronid |
| 13 | 17.7 | 435, 303 | 270, 330 | Quercetin arabinosid |
| 14 | 18.1 | 449, 433, 303 | 270, 330 | Quercetin ramnosid (Quercitrin) |
| 15 | 21.9 | 303 | 260, 370 | Quercetin |
| 16 | 23.4 | 317 | 260, 370 | Isorhamnetin |
| 17 | 24.3 | 287 | 240, 330 | Luteolin |
| 18 | 25.2 | 271 | 240, 320 | Apigenin |

**Table 3.** Analytical performances of the method used for the quantification of individual flavonoid compounds in buckwheat seeds, and after 3 and 7 days of sprouting.

| Validation Parameters | Calibration Equation | $R^2$ | LOD µg/g | LOQµg/g | Precision (RSD, %) | | Repeatability (RSD, %) |
|---|---|---|---|---|---|---|---|
| | | | | | Intra-Day | Inter-Day | |
| Hidroxi-secolariciresinol | Y = 11307X + 4493 | 0.992 | 10.40 | 31.52 | 3.15 | 2.85 | 4.31 |
| Epigallocatechin | Y = 11198X + 2849 | 0.993 | 9.65 | 29.24 | 2.49 | 2.62 | 3.56 |
| Epigallocatechingallat | Y = 3240X − 3511 | 0.990 | 16.84 | 51.03 | 2.36 | 4.1 | 2.82 |
| Gallocatechingallat | Y = 8671X + 2133 | 0.994 | 18.21 | 55.18 | 1.89 | 2.36 | 4.27 |
| Coumestrol | Y = 4735X − 2771 | 0.995 | 0.82 | 2.48 | 2.23 | 3.04 | 3.95 |
| Catechin | Y = 1435X + 305 | 0.993 | 0.35 | 1.06 | 2.81 | 5.01 | 3.97 |
| Secolariciresinol | Y = 1001X − 261 | 0.998 | 1.65 | 5.00 | 2.65 | 3.25 | 4.04 |
| Orientin | Y = 5100X − 4382 | 0.998 | 2.31 | 7.00 | 3.45 | 3.89 | 5.01 |
| Isoorientin | Y = 1294X + 525 | 0.991 | 1.15 | 3.48 | 4.25 | 3.87 | 3.26 |
| Vitexin | Y = 7555X + 2105 | 0.998 | 2.47 | 7.48 | 3.29 | 4.32 | 3.57 |
| Rutin | Y = 3022X + 156 | 0.996 | 11.02 | 33.39 | 2.47 | 2.65 | 2.12 |
| Isorhamnetin glucuronid | Y = 43279X − 1900 | 0.996 | 3.45 | 10.45 | 2.43 | 3.16 | 5.04 |
| Quercetin arabinosid | Y = 74639X + 2708 | 0.994 | 8.31 | 25.18 | 1.26 | 2.08 | 3.44 |

**Table 3.** *Cont.*

| Validation Parameters | Calibration Equation | R² | LOD µg/g | LOQ µg/g | Precision (RSD, %) | | Repeatability (RSD, %) |
|---|---|---|---|---|---|---|---|
| | | | | | Intra-Day | Inter-Day | |
| Quercetin ramnosid | Y = 54104X − 5319 | 0.991 | 9.03 | 27.36 | 3.11 | 3.15 | 2.27 |
| Quercetin | Y = 53615X − 1292 | 0.992 | 0.32 | 0.97 | 4.48 | 1.58 | 32.23 |
| Isorhamnetin | Y = 69535X − 2208 | 0.993 | 10.5 | 31.82 | 3.04 | 4.28 | 4.98 |
| Luteolin | Y = 14473X − 4389 | 0.995 | 4.10 | 12.42 | 2.97 | 3.18 | 3.61 |
| Apigenin | Y = 24574X + 5685 | 0.995 | 6.02 | 18.24 | 3.65 | 3.78 | 3.18 |

R²—correlation coefficient; LOD—detection limit; LOQ quantification limit; RSD—relative standard deviation.

**Table 4.** The concentrations of flavonoids extracted, in µg/g dw, in buckwheat achenes, and buckwheat achenes after 3 and 7 days of sprouting.

| No. | Compound | Buckwheat's Achenes, µg/g dwb | Buckwheat's Achenes after 3 Days of Sprouting, µg/dwb | Buckwheat's Achenes after 7 Days of Sprouting, µg/g dwb |
|---|---|---|---|---|
| 1 | Hydroxysecoisolariciresinol | 886.70 ± 19.02 | 604.52 ± 25.01 | 145.52 ± 4.88 |
| 2 | EGC | 131.58 ± 15.22 | 28.22 ± 6.22 | 85.55 ± 5.33 |
| 3 | EGCG | 44.20 ± 5.24 | 88.40 ± 10.39 | n.d. |
| 4 | GCG | 56.13 ± 6.66 | 32.47 ± 8.77 | n.d. |
| 5 | Coumestrol | 34.47 ± 4.87 | 395.23 ± 25.1 | 917.71 ± 15.74 |
| 6 | Catechin | 108.97 ± 5.21 | 139.49 ± 7.46 | 170 ± 8.54 |
| 7 | Secoisolariciresinol | 323.51 ± 6.98 | 229.67 ± 10.85 | n.d. |
| 8 | Orientin | 178.93 ± 7.09 | 230.28 ± 9.56 | 2361.50 ± 24.99 |
| 9 | Isoorientin | 114.28 ± 8.04 | 342.86 ± 9.34 | 1475.85 ± 35.88 |
| 10 | Vitexin | 155.74 ± 6.79 | 173.22 ± 12.40 | 5115.31 ± 37.65 |
| 11 | Rutin | 37.52 ± 3.77 | 420.47 ± 13.56 | 558.47 ± 8.22 |
| 12 | Isorhamnetin glucuronid | 39.23 ± 6.85 | 326.15 ± 9.91 | 385.17 ± 7.35 |
| 13 | Quercetin arabinosid | 299.19 ± 24.02 | 713.36 ± 27.44 | 885.17 ± 16.43 |
| 14 | Quercetin ramnosid (Quercitrin) | 227.04 ± 16.54 | 547.38 ± 12.74 | n.d. |
| 15 | Quercetin | 73.37 ± 3.63 | 173.31 ± 8.87 | 279.75 ± 9.58 |
| 16 | Isorhamnetin | 130.83 ± 5.44 | 183.03 ± 11.57 | 618.17 ± 8.33 |
| 17 | Luteolin | 52.02 ± 3.44 | 260.13 ± 8.89 | 184.91 ± 4.75 |
| 18 | Apigenin | 54.58 ± 3.98 | 111.21 ± 12.47 | n.d. |

In Table 4 and Figure 3 are quantified the flavonoids present in buckwheat achenes, and in sprouted buckwheat after three and seven days.

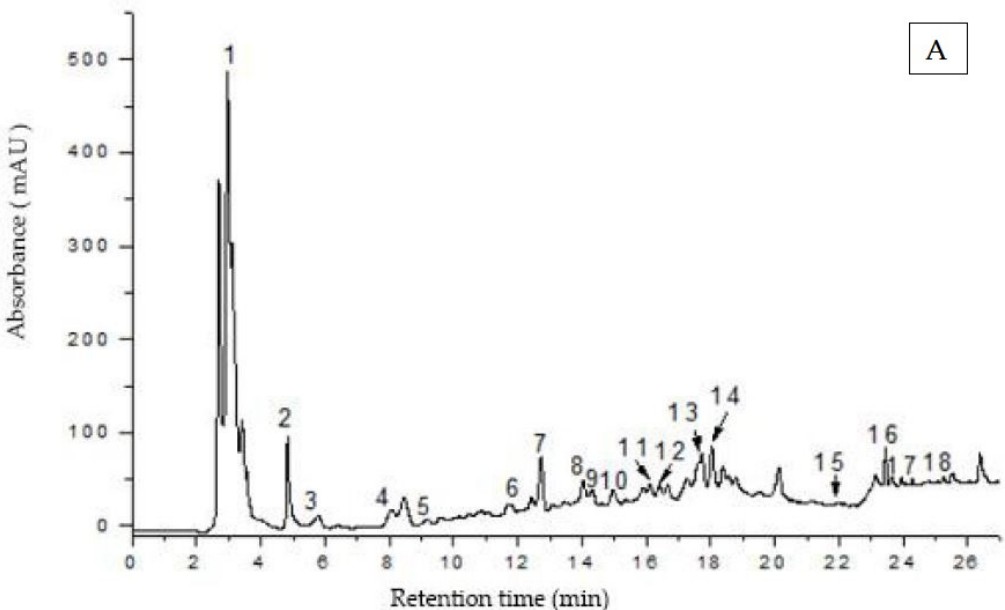

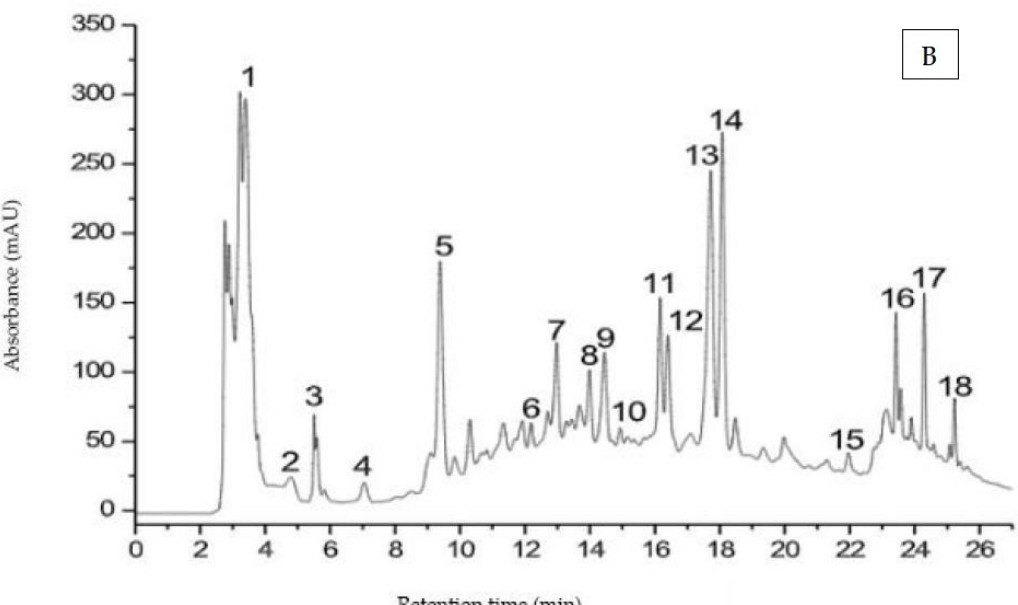

**Figure 3.** *Cont.*

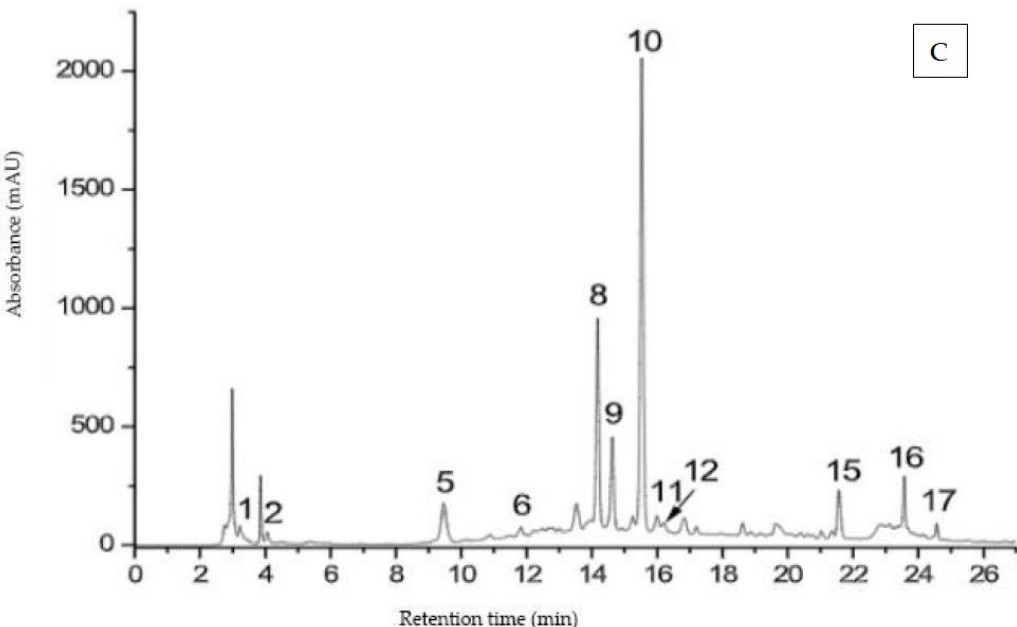

**Figure 3.** HPLC chromatogram of phenolic compounds in buckwheat achenes (**A**), buckwheat sprouted for 3 days (**B**) and buckwheat sprouted for 7 days (**C**). 1—Hydroxyseisocolariciresinol; 2—Epigallocatechin (EGC); 3—Epigallocatechingallat (EGCG); 4—Gallocatechingallate (GCG); 5—Coumestrol; 6—Catechin; 7—Secoisolariciresinol; 8—Orientin; 9—Isoorientin; 10—Vitexin; 11—Rutin; 12—Isorhamnetin glucuronide; 13—Quercetin arabinoside; 14—Quercetin ramnoside (Quercitrin); 15—Quercetin; 16—Isorhamnetin; 17—Luteolin; 18—Apigenin.

For quantitative analysis, the HPLC response, taken as the peak area under curve, was plotted against standard concentrations.

It is observed that in buckwheat achenes (Table 4, Figure 3), there are appreciable quantities of three C-glycosyl flavones: vitexin (apigenin-8-C-glucoside), orientin (luteolin 8-C-glucoside) and iso-orientin (luteolin 6-C-glucoside). Studies have shown that in other buckwheat species, for example, *Fagopyrum tataricum* and *Fagopyrum cimosum*, these flavones are in much smaller quantities [20,33] In the present study, these three flavones were found in significant quantities compared to the common buckwheat species analyzed by Kim and Colab [34] (undetectable). After three days, slight increases were observed, higher than in the study conducted by Kim and colleagues [34], but smaller than in the study conducted by Koyama and colleagues [21] (isoorientin: 0.8 mg/g, orientin: 1.6 mg/g, isovitexin: 3.6 mg/g and vitexin: 3.9 mg/g of extracts). After seven days, in our study, there were pronounced increases. Thus, vitexin increased by 33 times, while orientin and isoorientin increased by 13 times after 7 days of sprouting. Research has shown that orientin has a vasodepressant effect, isoorientin is a vasorelaxant, and vitexin has antibacterial action [20].

The other two flavones present in buckwheat achenes—apigenin and luteolin—were present in similar amounts of 54 μg/g DW. After three days of sprouting, the amount of lutein increased by five times, and apigenin by two times. After seven days there was a halving of the amount of luteolin and undetectable amounts of apigenin. These two flavones have antioxidant and anticancer properties [34].

Rutin, one of the most important flavonol glycosides—recognized for its antioxidant, anticancer [30] antibacterial [33–35] antifungal [36,37], neuroprotective [38], vasoprotective [39] anticonvulsant [40], peripheral and central antinociceptive [41] and anti-inflammatory activities [42,43]—is a selective and non-toxic modulator of hypercholesterolemia, [44] an antihypertensive [45] and a possible stimulant of osteoblasts [46]; it was found in larger quantities in young green leaves—especially in flowers—and also in small quantities in achenes [47]. In the achenes of our buckwheat specie a lower

quantity was observed, compared with other common buckwheat species studied (about 135.5 μg/g dw) [20]. However, during sprouting, it increased by 11 times after 3 days, and by 15 times after 7 days (Table 4), reaching a content of 558.47 μg/g dw. It has been shown that the rutin content of leaves, not flowers, can vary during sprouting, depending on the level of intensity of UV radiation—being lower when the radiation intensity is lower [48]. For example, Lee et al. [49] recorded a 19-fold increase in rutin content after 7 days of sprouting. In buckwheat achenes, the highest concentrations of flavonols were represented by quercetin arabinoside (299.19 μg/g sample) and quercitrin (quercetin-3-O-rhamnoside; 227.04 μg/g dw)—much higher amounts compared to the variety of buckwheat (undetectable) from the study conducted by Kim et al. [50]. From Table 4 and Figure 3, it can be seen that the quercitrin content reached a maximum on the third day, after which it decreased, so that on the seventh day it was undetectable. The quercetin content showed a 2.3-fold increase after 3 days of sprouting, compared to the results obtained by Zhang et al. (increased by 1.61 times) [8]. After 7 days of sprouting, there was an increase of 3.8 times. Knowing the numerous pharmacological actions of quercetin [51], it is possible to assess the maximum level of quercetin content in natural products so that it can be used successfully in various diseases; for example, in recent studies that highlight its antiviral action in SARS-Cov2 infections [52–54].

Even in smaller quantities, the flavonols isorhamnetin (130.83 μg/g dwb) and isorhamnetin glucuronide (39.23 μg/g dwb) showed significant increases after seven days of sprouting—by five times (isorhamnetin) and ten times (glucuronide), respectively. The identification and quantification of these two flavonols is important in terms of the multiple pharmacological effects that they have: antioxidative, anti-atherosclerotic, myocardial and endothelial protective [55–60].

In buckwheat achenes there were three flavanols identified: EGC, EGCG and GCG, whose concentrations gradually decreased, so that on the seventh day of sprouting, only 85.55 μg/g dwb of EGC remained. Rhubarb is from the same family as buckwheat, and contains 3.3 μg/g dwb epicatechin [61]. Hanefeld and Herrmann [62] showed in their study a decrease in epicatechin content in rhubarb stems and leaves during plant development. In contrast, the concentration of catechins from 108.97 μg/g dwb in buckwheat achenes increased to 170 μg/g dwb in sprouted buckwheat. The same evolution of catechin concentration has been previously reported by Benincasa et al. [63].

Studies have shown that the antioxidant activity given by flavanols would be higher than that given rutin [64], and lower than that given by quercetin—although it works for a longer period of time.

Lignans belong to the group of diphenolic compounds derived from the biosynthetic pathway of shikimic acid [65]. In buckwheat achenes there were secoisolariciresinol and hydroxisecoisolariciresinol, identified in quantities of 323.51 μg/g dwb and 886.70 μg/g dwb, respectively—compounds with antiestrogenic, antioxidant and anticarcinogenic properties [65]. Coumestrol, a natural organic compound of the coumestans class, is classified as a phytoestrogen because it has actions similar to those of estradiol [66]. In buckwheat achenes we determined a concentration of 34.47 μg/g dwb. After 7 days of sprouting, its concentration increased by 27-fold, reaching 917.71 55 μg/g dwb.

### 3.4. FTIR Analysis

According to the fundamentals of IR spectroscopy, there are 3 absorption domains (Figure 4): 4000–2500 cm$^{-1}$ (I), 2000–1500 cm$^{-1}$ (II) and the fingerprint region (1500–400 cm$^{-1}$) [67]. In this study, we performed the first spectroscopic characterization of buckwheat achenes sprouted after three and seven days. The spectral domains most often used in the qualitative analysis of buckwheat achenes are those in region I, characteristic of O–H bonds, in the composition of phenols [68]. Thus, in the unsprouted buckwheat achenes (Figure 4) the wide band from 3273.8 cm$^{-1}$ was observed. In the sprouted achenes after 3 days, there was an increase in the transmittance to 3278 cm$^{-1}$ due

to the accumulation of -OH groups in the phenol structures, showing accentuated growth after 7 days.

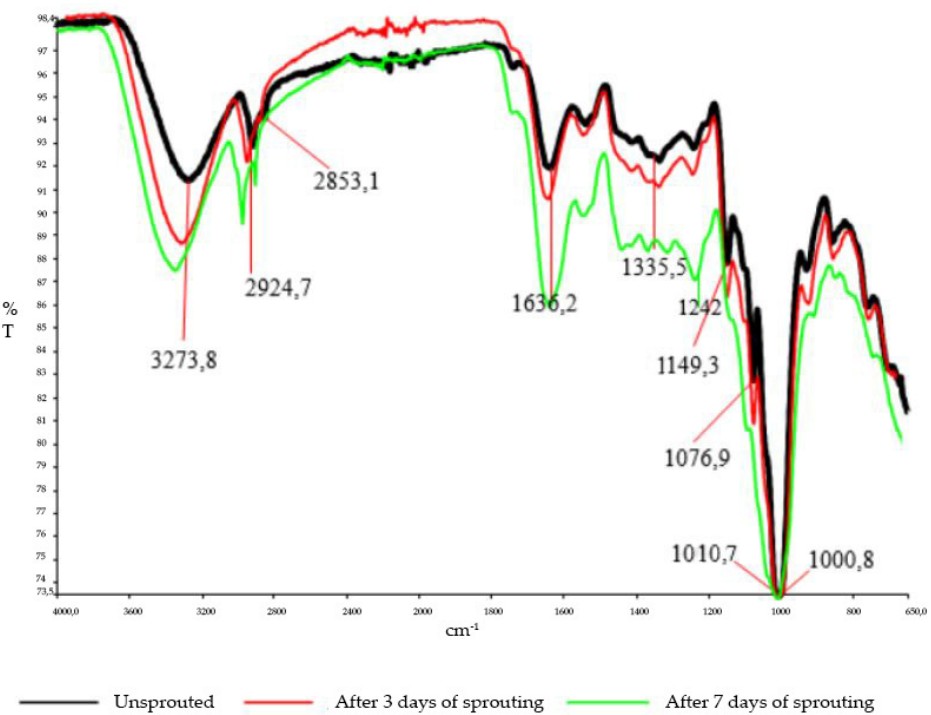

**Figure 4.** FTIR footprint of bioactive compounds in buckwheat achenes.

Adjacent peaks at 2924 and 2853.1 $cm^{-1}$ corresponded to the tensile vibrations of the -C–H bond within the -CH$_3$ or -CH$_2$ groups [69]. Region II showed a considerable increase due to the tensile vibrations in the carbonyl groups, as a result of the accumulation of carboxylic acids and amino acids.

In summary, our experimental studies have shown that the analyzed Moldavian buckwheat extracts contain important lignans, flavonoids and isoflavonoids (Figure 5), which have known beneficial effects such as antioxidant, anti-inflammatory, antiallergic, antiestrogenic, antiviral and anticarcinogenic properties, and can be recommended for exploration in functional food due to their protective effects.

In the long run, understanding the effects of food ingredients on human health will lead to a change in the way we eat and thus a decrease in the risk of morbidity and mortality due, in particular, to obesity and cardiovascular disease. Both expanding the range and the insight into the role of functional foods would lead to easier choices for consumers.

**Figure 5.** Main compounds of analyzed buckwheat extracts and their biologic activities.

## 4. Conclusions

In this study the concentration of different representatives of flavonoids was analyzed, from the subclasses: flavones, flavonols, flavan-3-oils, isoflavones and flavanones. The results showed a significant number of flavonoids in buckwheat achenes. After three days of sprouting, we recorded the highest increases in flavonoids, represented by rutin but also by quercetin, quercetin-arabinoside, lutein and apigenin. After seven days of sprouting, the highest content was for vitexin, followed by orientin. The flavonoid content of both buckwheat achenes and sprouted buckwheat, after three and seven days, increased continuously, which caused a higher antioxidant activity, expressed as DDPH. FTIR analysis revealed the accumulation of -OH functional groups characteristic of phenolic compounds. From an economic point of view, it is useful to identify the optimal time for obtaining significant quantities of these flavonoids.

Our studies demonstrate once again that both buckwheat achenes and sprouted buckwheat, after three and seven days, can be used as food, but also as ingredients for the development of other functional foods or for producing pharmaceutical preparations.

**Author Contributions:** Conceptualization, C.D. and C.V.; methodology, C.V., G.-E.B, R.M.D.; software, L.B.; validation, M.D. and L.B.; formal analysis, A.V.I.; investigation, C.D., A.V.I., G.-E.B. and R.M.D.; resources, A.V.I.; data curation, M.D.; writing—original draft preparation, C.D.; writing—review and editing, C.D., C.V. and R.M.D.; visualization, M.D. and L.B.; supervision, C.D., C.V. and R.M.D.; project administration, G.-E.B.; funding acquisition, R.M.D., L.B., A.V.I. All authors have read and agreed to the published version of the manuscript.

**Funding:** This research received no external funding.

**Institutional Review Board Statement:** Not applicable.

**Informed Consent Statement:** Not applicable.

**Data Availability Statement:** Not applicable.

**Conflicts of Interest:** The authors declare no conflict of interest.

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
