# Peer review of "New Insights into the Antioxidant Compounds of Achenes and Sprouted Buckwheat Cultivated in the Republic of Moldova"

_applsci, doi:10.3390/app112110230_

Round 1

Reviewer 1 Report

Reviewer's comments :

The title of the paper should be improved - suggests that different varieties of buckwheat will be tested, but this is not the case. In addition, sprouts are not grown.

Line 124-125 - it should be clarified whether a specific variety of buckwheat was tested or whether the sample was a commercial mix

Line 126 - Chemicals would be more appropriate for this chapter

Line 141 - What was the purpose of sampling each day if seed samples were tested before germination and after 3 and 7 days of germination? Figure 6 is not included in the article

Line 190 - Shouldn't the formula look like this:

[(Acontrol – Asample)/ Acontrol]x100

Line 218-220 – The chapter Statistics does not describe all the statistical methods used, e.g. the correlation calculation methods discussed in chapter 2.3. What method was used to determine the significance of differences between the mean values. In the discussion of the results, the Authors write about the significance of the differences in mean values, but it is not known on what basis.

Line 235 - in table 1, in row 5 of the Authors column, there is an unexplained reference (1). I suppose it should be [0]?

Line 236 - it is 2.3 .. it should be 2.3.

Line 237-245 - The text contains an overview of the results not presented in the publication in any table or figure. The data should be illustrated.

Line 248-249 - The sentence should be included in chapter 2.1 or 2.4

Line 255 – Table 2 - It is worth assigning the compounds marked in Table 2 to the following groups: flavones, flavonols, flavan-3 oils, isoflavones, flavanones

Line 258 – Is the cross-reference to Table 4 correct?

Line 260- Table 3

The column headings contain abbreviations and names of features that are not explained neither in chapter 2.8 nor in the reference below the table. What is LOD, LOQ, Precision intra- and inter day and Repetability? How were Calibration equation and R determined? No description in the Methodology. Different fonts have been used in this table should be standardized.

Line 268 - Figure 3 is difficult to read

Line 299 - an unnecessary hyphen in the word hypercholesterolemia

Line 343 - publications are written impersonally, and besides, this publication has several authors, so not "I".

Line 369-373 - A summary should be found in the next chapter. Figure 5 is redundant because it does not fully illustrate the results obtained by the authors of the publication. The source data for this drawing is also missing.

Line 377 - publications are written impersonally, and besides, this publication has several authors, so not "I".

Line 379 – “increased amounts of flavonoids” (?) - a large/signifficant amount of flavonoids

Line 382-383 – „Both buckwheat seeds and sprouted buckwheat, after 3 and 7 days, have increased continuously, which caused the antioxidant activity, expressed as DDPH, to be higher” - very imprecise term. I guess the point was that the increase in polyphenols during seed germination increased the antioxidant activity

Line 385-387 – “Knowing the concentrations of different representatives of flavonoids, makes it possible to know the optimal time to obtain, in a short time, which is economically profitable.” - What did the Authors mean? What "to obtain"?

Author Response

Thank you for your comments, please see attachment.

Reviewer 2 Report

The article deals with an interesting topic, but in my opinion, in its present form it cannot be published in Applied Sciences. Below You can find comments and recommendations:

  1. The article deals with 3 samples only. Statistical analysis is incomplete and should be improved. The mean values with standard deviations are not enough to compare and discuss the results - homogeous groups should be defined.
  2. line 50 - it should be: "are" instead of "is"
  3. line 61 - the sentence is incomplete and should be rearranged
  4. lines 70-72 - the style of the sentence is incorrect and should be rearranged.
  5. lines 77-78 - the style of the sentence is incorrect and should be rearranged.
  6. lines 81-82 - should the process of sprouting of seeds be classified as food processing procedure?
  7. lines 117-120 - the information should be rather in discussion chapter, not in the aim of the study.
  8. line 141 - there is no figure 6 in the text.
  9. line 234 - what do the Authors mena by [0]
  10. lines 257-258 - the sentence is incorrect and should be rearranged.
  11. the abbreviations are not listrd and explained below the tables.
  12. figure 3 is presented incorrect.
  13. lines 295-302 - the information does not belong to a discussion section, it should be in an introduction section.
  14. lines 303-304 - [20] is given 2-times - it is not correct.
  15. lines 369-373 and. Fig. 5 - the information does not belong to a discussion section, it should be in an introduction section.
  16. Discussion and conclusions are poorly written and should be rearranged.

In my opinion the article has serious flaws and is not suitable to be published in Applied Sciences

Author Response

Thank you for your comments, please see attachment

Reviewer 3 Report

Comments on the manuscript ID1410460 by Dumitru et al. entitled “Comparative phytochemical profile of buckwheat seeds and sprouts grown in the Republic of Moldova”:

The authors report an experimental study of the phytochemical composition and antioxidant activity of buckwheat seeds and sprouts grown on the territory of Moldova. In general, the article is full of content material and beneficial information, but it requires some improvement in the presentation/discussion of the results. Some shortcomings are listed below:

  • Authors’ affiliation: University from Galati - University of Galati;
  • Abstract, line 23: the % increase of TPC and TFC must be rounded to the nearest whole number;
  • 1, line 41: classified into – classified as;
  • 1, lines 42-43: Flavonoids have 15-carbon skeleton, which consists of two phenyl rings and a heterocyclic ring, not of “2 benzene rings connected by a “bridge””;
  • Figure 1: all the structures presented on the figure are A-ring dihydroxy substituted – the basic structure of flavonoids (and of subclasses) is without OH-substitution at ring A;
  • 3, lines 124-125: more details for the seeds should be given; the use of seeds in should comply with international and national guidelines for the use of plant seeds in the study;
  • DPPH free radical scavenging is an accepted mechanism for screening the antioxidant activity, but it is important to note that DPPH test measures radical scavenging activity, not the antioxidant one;
  • Table 1: ref. (1)?
  • Table 3: the names Hidroxisecolariciresinol and Secolariciresinol are incorrect, the authors should check all names listed in table 3;
  • 10, line 289: avasorelaxant?;
  • 11, line 343; p.13, line 377: the authors should avoid first person pronouns (or at least to use “we”);
  • The last sentence makes no sense: “Knowing the concentrations of different representatives of flavonoids, makes it possible to know the optimal time to obtain, in a short time, which is economically profitable.”

Author Response

(The authors gave the same response as above.)
